



# A muographic study of a scoria cone from 11
# directions using nuclear emulsion cloud chambers
Seigo Miyamoto[1], Shogo Nagahara[1,2], Kunihiro Morishima[3], Toshiyuki Nakano[3],
Masato Koyama[4], Yusuke Suzuki[5]
[1]Earthquake Research Institute, The University of Tokyo, 1-1-1 Yayoi, Bunkyo-ku,
Tokyo, 113-0032, Japan.
[2]Graduate School of Human Development and Environment, Kobe University, 3-11
Tsurukabuto, Nada-ku, Kobe, Hyogo, 657-8501, Japan.
[3]Fundamental Particle Physics Laboratory, Graduate School of Science of Nagoya
University, Furocho, Chikusa-ku, Nagoya, Aichi, 464-8602, Japan.
[4]Department of Education, Shizuoka University, 836 Ohya, Suruga-ku, Shizuoka City,
Shizuoka, Japan.
[5]STORY, Ltd., 2-2-5-2321, Minatomachi, Naniwa-ku, Osaka City, Osaka, Japan.
*Correspondence to*: Seigo Miyamoto (miyamoto@eri.u-tokyo.ac.jp)



**Abstract**
One of the key challenges for muographic studies is to reveal the detailed 3D density
structure of a volcano by increasing the number of observation directions. 3D density
imaging by multi-directional muography requires that the individual differences in the
performance of the installed muon detectors are small and that the results from each
detector can be derived without any bias in the data analysis. Here we describe a pilot
muographic study of the Izu–Omuroyama scoria cone in Shizuoka Prefecture, Japan,
from 11 directions, using a new nuclear emulsion detector design optimized for quick
installation in the field. We describe the details of the data analysis and present a
validation of the results.
The Izu–Omuroyama scoria cone is an ideal target for the first multi-directional
muographic study, given its expected internal density structure and the topography
around the cone. We optimized the design of the nuclear emulsion detector for rapid
installation at multiple observation sites in the field, and installed these at 11 sites
around the volcano. The images in the developed emulsion films were digitized into
segmented tracks with a high-speed automated readout system. The muon tracks in
each emulsion detector were then reconstructed. After the track selection, including
straightness filtering, the detection efficiency of the muons was estimated. Finally, the
density distributions in 2D angular space were derived for each observation site by
using a muon flux and attenuation models.
The observed muon flux was compared with the expected value in the free sky, and is
88% ± 4% in the forward direction and 92% ± 2% in the backward direction. The
density values were validated by comparison with the values obtained from gravity
measurements, and are broadly consistent, except for one site. The excess density at





this one site may indicate that the density inside the cone is non-axisymmetric, which
is consistent with a previous geological study.
**1 Introduction**
Scoria or cinder cones are a simple volcanic structure, along with stratovolcanoes,
shield volcanoes, and lava domes. Understanding the internal structure of scoria cones
is important for volcanic hazard assessments. The internal structure of scoria cones
has been mainly investigated by geological approaches. Kereszturi and Németh (2012)
presented a schematic cross-section of typical scoria cones, and Geshi and Neri (2014)
presented detailed photographs of the feeder dike and interior of a scoria cone formed
by the 1809 Etna eruption. Yamamoto (2003) investigated outcrops of the interior of
scoria cones in the Ojika-jima monogenetic volcano group, Nagasaki Prefecture, Japan.
Yamamoto (2003) classified 40 scoria cones according to their degree of interior welding
and proposed a link between lava outflow and cone collapse. However, scoria cones with
such outcrops are rare, and the internal structure can vary markedly among cones.
Therefore, non-destructive methods are required to investigate scoria cones that lack
outcrops.
Muography is a non-destructive technique for investigating the internal density
structure of large objects, employing the strong penetrating force of muons, which are
high-energy elementary particles contained in cosmic rays. Muography has also been
used for studying volcanoes, including visualization of a shallow conduit (e.g., Tanaka
et al., 2009), detection of temporal changes in water level due to hydrothermal activity
(Jourde et al., 2016), and 3D density imaging of a lava dome using a joint inversion of
muographic and gravity data (Nishiyama et al., 2017).



64 In unidirectional muography, the only measurable quantity is the density length,

65 which is the integral of density and length along the muon direction. It has no spatial

66 resolution along the muon path. Therefore, even if an interesting density contrast is

67 found below the crater, this could reflect contributions from other parts of the volcanic

68 body. Similar to X-ray computed tomography, which has been developed as a 3D density

69 imaging technique, muography can obtain 3D spatial resolution by increasing the

70 number of observation directions. In previous studies, muography of volcanoes has

71 been conducted in two or three directions (Tanaka et al., 2010; Rosas-Carbajal et al.,

72 2017). However, the spatial resolution is not sufficient to determine the detailed

73 structure of the volcanic interior. Nagahara and Miyamoto (2018) undertook a 3D

74 density reconstruction based on multi-directional muography and the filtered back-

75 projection technique. Their study showed that it is necessary to increase the number of

76 directions to obtain 3D spatial resolution in volcanological studies.

77 Nuclear emulsion is a type of muon detector, and has been used for studies of

78 volcanoes (Tanaka et al., 2007; Nishiyama et al., 2014; Tioukov et al., 2019). The

79 trajectories of high-energy charged particles that pass through an emulsion film are

80 recorded as aligned silver grains with micron-scale resolution (Nakamura et al., 2005;

81 Tioukov et al., 2019; Nishio et al., 2020). The positions and slopes of aligned grains in a

82 developed emulsion film are digitized with an automated emulsion readout system

83 (Kreslo et al., 2008; Morishima and Nakano, 2010; Bozza et al., 2012; Yoshimoto et al.,

84 2017). Unlike hodoscopes using scintillator bars (e.g., Saracino et al., 2017) or multi-

85 wire proportional chambers (Olah et al., 2018), a nuclear emulsion film does not have

86 temporal resolution. In contrast, an emulsion detector does not require electricity,

which facilitates the installation of such detectors around volcanoes where the
infrastructure is not well developed.
In muographic studies of a volcano, contamination by low-momentum particles must
be removed to derive the correct density (Nishiyama et al., 2014, 2016). Thus, nuclear
emulsion detectors have often been used as an emulsion cloud chamber (ECC), which
comprises alternating layers of films and lead or iron plates (e.g., Kodama et al., 2003).
An ECC detector can measure the momentum of the charged particle by detecting
deflection angles caused by multiple Coulomb scattering (Agafonova et al., 2012). For
multiple Coulomb scattering, there is a relationship between the maximum detectable
momentum $p_{max}$ and position resolution $y_{reso}$ as follows:
$$p_{max} \sim \alpha\, X_0^{-0.5}\, x^{1.5}\, y_{reso}^{-1} \qquad\qquad (1)$$
where $\alpha$ is a constant, $X_0$ is the radiation length of a material, and $x$ is the thickness of
the material. The position resolution of the newest scintillator hodoscope or MWPC is
on the order of 1 mm (Saracino et al., 2017; Olah et al., 2018). In the case of nuclear
emulsion, the resolution is about 1 μm. When using ECC, the thickness of the material
can be reduced to 1/100 while maintaining the same $p_{max}$, which is advantageous in
terms of transportation in the field.
A new design of the ECC detector was also required for its rapid installation at
multiple observation sites in the field. In a previous study of volcano observations
using the ECC detector (Nishiyama et al., 2014), rapid installation of the detector was
not required because the number of observation sites was just one. It is also important
to establish a data analysis procedure for the muon tracks recorded by the ECC
detectors. To derive an accurate density value for the volcanic body, it is necessary to
remove low-momentum contamination, estimate the detection efficiency, and validate



the results. In addition, for bias-free 3D imaging by multi-directional muography, the
installed muon detectors must show similar performance.
**2 Izu–Omuroyama scoria cone**
The Izu–Omuroyama scoria cone (34°54'11"N, 139°05'40"E; 580 m a.s.l.) is one of the
largest scoria cones in the world, and is part of the Higashi Izu monogenetic volcano
group (Aramaki and Hamuro, 1977), which is located in the northeastern Izu
Peninsula, Ito City, Shizuoka Prefecture, Japan. It is considered to have formed at 4
ka, based on $^{14}$C dating (Saito et al., 2003). The basal diameter is 1,000 m, the height is
280 m from the base, and the typical slope of its flanks are 29–32°. The center of the
cone contains a crater that is 250 m wide and 40 m deep. The volume of the cone is
$71 \times 10^6$ m$^3$, and lava with a volume of $\sim 10^8$ m$^3$ has flowed out from the base of the
cone (Koyano et al., 1996). The lava is a basaltic andesite with 54–56 wt.% $SiO_2$
(Hamuro, 1985).
Although the shape of the Izu–Omuroyama scoria cone appears to be axisymmetric
(Fig. 1), a geological study suggested it has an anisotropic structure due to the
following reasons. (i) During/after the growth of the cone, some interior parts became
welded due to loading, residual heat, and a low cooling rate. As a result, some denser
material formed. (ii) At the end of the eruption, a lava lake was formed in the crater,
and the lava flowed out to the western foot of the cone. (iii) There is a small crater on
the south side of the cone, which is thought to have formed when the main crater was
blocked at the end of the eruption (Koyano et al., 1996).
The bulk density of typical continental crust is about $2.6$–$2.7 \times 10^3$ kg m$^{-3}$. The bulk
densities reported for scoria deposits are $0.84$–$1.01 \times 10^3$ kg m$^{-3}$ (Taha and Mohamed,



2013) and $0.56$–$1.20 \times 10^3$ kg m$^{-3}$ (Bush, 2001). Therefore, the maximum expected
density contrast is about $1.4$–$2.0 \times 10^3$ kg m$^{-3}$, due to the difference in porosity
between welded rocks and scoria deposits. In addition, the Izu–Omuroyama scoria cone
is an ideal target for multi-directional muography due to the accessibility to detector
sites and absence of muographic shadows from any direction caused by other
topographic features.





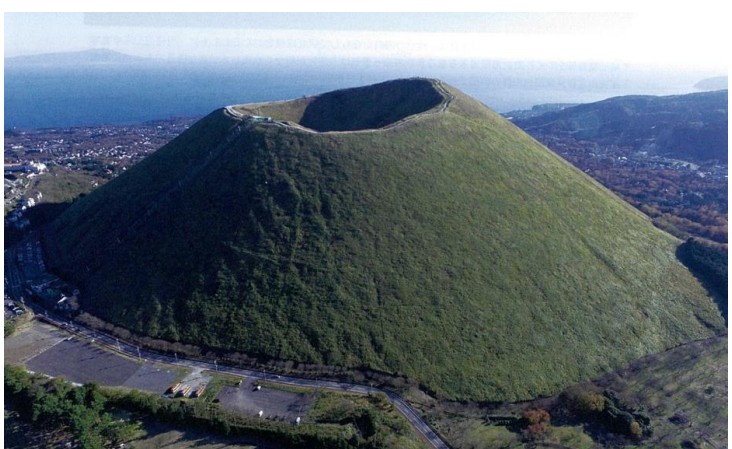

Figure 1. Photograph of the Izu–Omuroyama scoria cone from the northwest, taken by
an unmanned aerial vehicle (Koyama, 2017).



**3 Multi-directional muography observations using emulsion cloud chambers**
**3.1 Detector design**
Emulsion films were manufactured by pouring 70 µm of nuclear emulsion on both
sides of a 180 µm-thick plastic base. The size of a film is $125 \times 100$ mm. The films were
vacuum-packed in a light-blocking envelope to maintain their planar form, which
prevented air bubbles forming between the envelope and film, and made it easy to
handle the films in the field.
The detector used for the 2018 observations is basically the same as that of Nishiyama
et al. (2014), and only the number of lead plates was different. The former consists of
20 films and 9 plates of 1-mm-thick lead, the latter consists of 20 films and 19 lead
plates. At the time of installation in 2018, the films, lead plates, and supports were all
in pieces and, therefore, a lot of time and effort was required for assembly in the field.
The more efficient detector design was required for rapid and error-free installation.
The detector used in the 2019 observations was improved. It consists of an ECC and
an outer box. The ECC consists of 20 emulsion films and 19 lead plates, each 1 mm
thick (Fig. 2a). An aluminum frame was fixed to a lead plate with a thin sheet of glue,
and then an emulsion film with the light-blocking envelope was attached with scotch
tape. In this paper, we term this unit the emulsion–lead plate (EL plate; Fig. 2a). The
EL plate was designed for quick assembly in the field.
The outer box consists of 10-mm-thick aluminum plates (Fig. 2b). The outer size of
this box is 190 mm in width, 155 mm in height, and 90 mm in depth. An ECC and
strong springs were placed in the box. There are four screw holes on one side of the box,
and by turning the bolts and pushing the spring plate, a uniform pressure ($\sim 10^5$ Pa)





was applied to the ECC. This pressure prevents the film from stretching and shrinking
due to temperature changes.
Given that there is no temporal resolution in emulsion films, we needed to add time
information to the ECC. In previous muographic studies using emulsion films,
researchers have used emulsion films with a different alignment during the muon
observations and standby (e.g., Tanaka et al., 2007). In the present study, the corners of
the EL plates were aligned during the muon observations, while the corners were
intentionally shifted a few millimeters horizontally and fixed with clamps during
standby (Fig. 3). This alignment difference distinguishes passing charged particles
during non-observation and observation periods by pattern matching of each emulsion
film. By using this procedure, the time to set the alignment between each EL plate in
the field is <30 s. Although the muon tracks that pass through an ECC during the
alignment set-up may become noise, our procedure reduced such tracks.



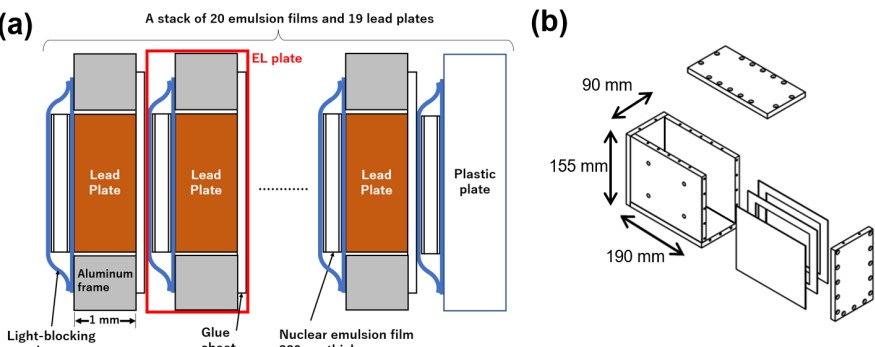

Figure 2. Design of the ECC and outer box. (a) Schematic cross-section of the EL plates and an ECC. The EL plate consists of a 1-mm-thick aluminum frame, 1-mm-thick lead plate, 100-μm-thick glue sheet that fixes a lead plate to an aluminum frame, and an emulsion film with a light-blocking envelope. An ECC consists of 19 EL plates and an emulsion film with a plastic plate. (b) Schematic of the aluminum outer box. The thickness of the aluminum plate is 10 mm. The ECC shown in (a) was set inside this box. There are four holes for feed screws in the front plate.






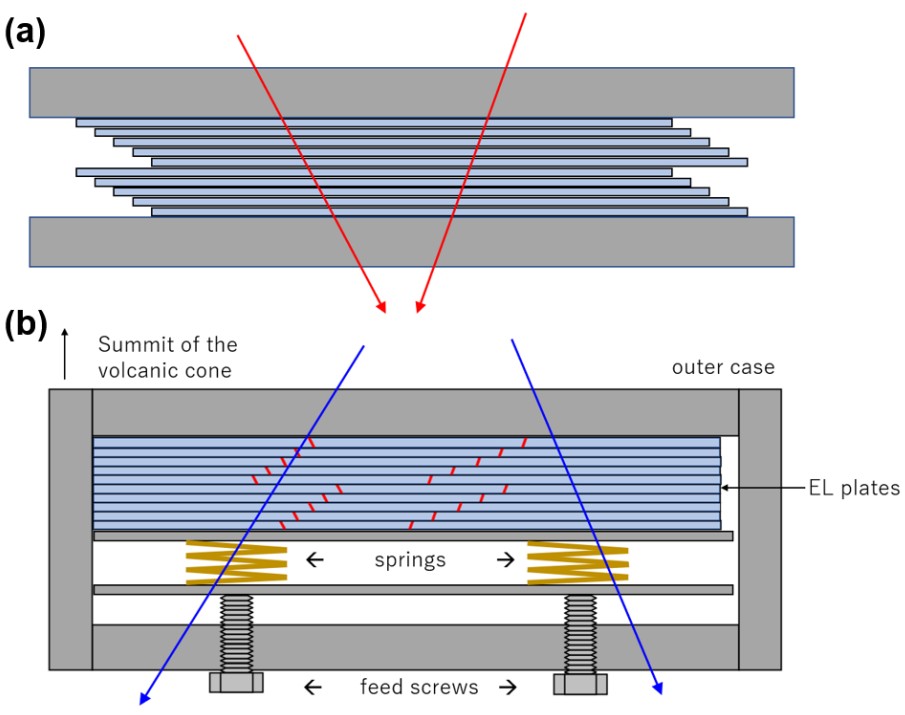



Figure 3. (a) View of the EL plates from above during standby. The EL plates were
intentionally shifted a few millimeters horizontally and fixed with a pair of steel plates
and clamp. The red lines represent the muon tracks in this alignment. (b) View from
above during the observations. The EL plates were aligned to the side of the outer box,
and fixed by the springs and feed screws. The blue lines represent muon tracks during
observations. Note that the red tracks cannot be reconstructed in this alignment.



### 3.2 Installation

The detectors were installed at three sites in 2018 and eight sites in 2019 around the Izu–Omuroyama scoria cone (Fig. 4; Table 1). Each detector was buried in a hole that was about 40 cm deep to avoid high temperatures due to direct sunlight. This is done because the number of latent image specks decreases, and the number of randomly generated specks increases, under high-temperature conditions (Nishio et al., 2020).






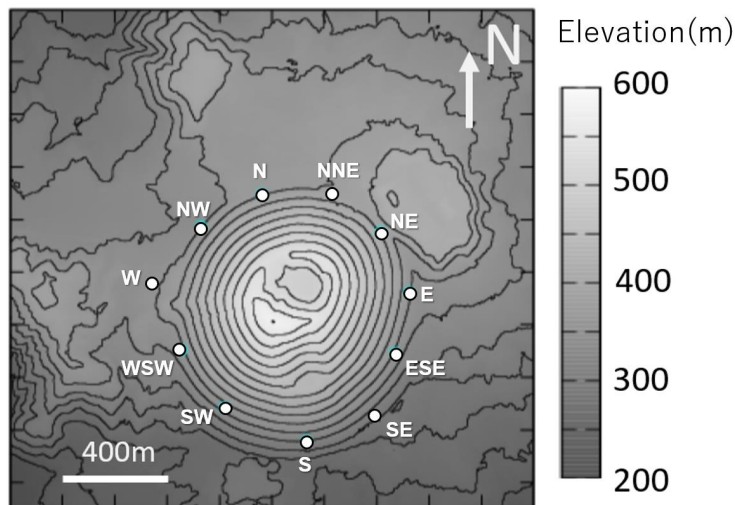



Figure 4. Topography of the Izu–Omuroyama scoria cone. White dots represent
observation sites.

| Detector site ID | Effective area (cm$^2$) | Exposure time (days) |
|---|---|---|
| W, SE, and NNE (2018) | 120 | 60 |
| N, NW, WSW, SW, S, ESE, E, and NE (2019) | 240 | 90 |

Table 1. Effective area and muon exposure time for each detector.




The installation procedure at each observation site in 2019 was as follows (Fig. 5).
1)  Carry the outer box and EL plates to the observation site.
2)  Measure the coordinates of the site with a hand-held GPS (GERMIN; model GPS
eTrex 30J). The typical uncertainty of the latitudinal and longitudinal coordinates
is 3 m.
3)  Dig a hole in the ground with horizontal dimensions of 60 × 40 cm and a depth of
40 cm.
4)  Flatten the base of the hole, place a plastic bag inside the hole, and lay down a
piece of plywood.
5)  Put double-sided tape on the bottom of the outer box and place it on the plywood.
6)  Put the stack of EL plates into the box and quickly align these (<30 s).
7)  Close the cap of the outer box.
8)  Turn the feed screws to increase the pressure.
9)  Measure the attitude of the outer box (i.e., the yaw [azimuth], roll, and pitch) with
a fiber optic gyro (Japan Aviation Electronics Industry Ltd.; model FOG JM7711;
Watanabe et al., 2000) and digital leveler. The typical errors on the yaw, roll, and
pitch are $8.7 \times 10^{-3}$, $1.0 \times 10^{-3}$, and $1.0 \times 10^{-3}$, respectively.
10) Cover with styrofoam to avoid heating from the ground surface.
11) Close the plastic bag to keep water out.
12) Backfill the hole.

The time taken for this installation was ~2 h for each site, and we installed detectors
as three sites in a day in 2019. The detector retrieval procedure was the opposite of the



installation procedure. The 380 films were developed in a darkroom. The deposited
silver particles on the surface of the films were removed with anhydrous ethanol. The
gelatin of the sensitive layer was swollen with a glycerin solution to obtain the
optimum thickness for an automated track readout system, which is described in the
next section.






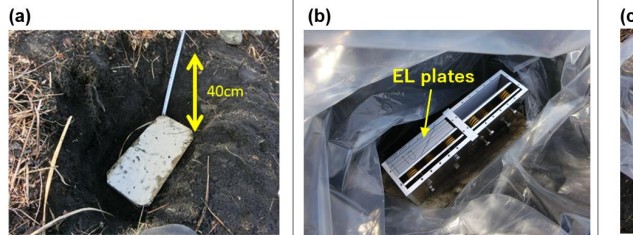



Figure 5. Photographs showing the installation procedure. (a) Dig a hole and place a
plywood sheet in the bottom. (b) Place the outer box in the hole and put a stack of EL
plates into the box. The plates were aligned over a period of <30 s. After closing the top
plate of the box, the feed screws were tightened to increase the pressure. (c) The yaw,
roll, and pitch were measured with a fiber optic gyro (FOG) and digital leveler.

**4 Track reconstruction, selection, and detection efficiency estimation**
**4.1 Track reconstruction**
A track of a high-energy charged particle is recorded as an aligned line of silver grains
in the emulsion film (e.g., Nakamura et al., 2005). The images in the 380 nuclear
emulsion films were scanned and the positions and slopes of the tracks were digitized
by "HTS", which is a high-speed automated track readout system at Nagoya University
(Yoshimoto et al., 2017). For each ECC, the tracks of the charged particles were
digitally reconstructed from the segmented tracks in 20 films. NETSCAN 2.0 software
was used for track reconstruction (Hamada et al., 2012). NETSCAN 2.0 rapidly
corrects for film distortions and local misalignments between films by using many
tracks recorded over a large area. It then outputs all possible connections as the final
result. NETSCAN 2.0 has been used in various fields, such as neutrino physics
(Hiramoto et al., 2020), cosmic ray astronomy (Takahashi et al., 2015), and muographic
studies of Egyptian pyramids (Morishima et al., 2017). The typical procedure for the
track reconstruction is as follows.

1)  Reconstruct the "base track", which is connected between the emulsion layers

across the plastic base of 170 μm in a film.

2)  Reconstruction of the "linklet", which is the base track pair between adjacent films

across lead plates.

3)  Reconstruction of the tracks that connect across the whole ECC. If no base track

was found in two consecutive films on the extension of a track, then the track was

considered to have stopped.




For example, in ECC_ID = 02, 8.9 $\times$ $10^6$ base tracks, 3.2 $\times$ $10^6$ linklets in a pair of
adjacent films, and 1.7 $\times$ $10^7$ tracks in an entire ECC were reconstructed.
**4.2 Track selection**
NETSCAN 2.0 outputs all possible track connections. Therefore, it is necessary to
carefully select the tracks for the muographic analysis. A schematic example of the
output tracks is shown in Fig. 6. Most of the branches can be considered to represent
contamination by fake base tracks caused by random noise, or the coincidental
occurrence of low-energy positrons/electrons on parallel slopes in the vicinity of the real
tracks (e.g., Fig. 6; cases 2 and 3). Some branches consist of a pair of straight tracks
with small closest distances and similar angles (Fig. 6; case 4). In this case, the two
tracks should be separated.
The following $\chi^2/ndf$ value was calculated for all tracks for the low momentum cut-
off:
$$\chi^2/ndf = \sum_m \left[ \left( \frac{\Delta\theta_R^m}{\sigma_R^m} \right)^2 + \left( \frac{\Delta\theta_L^m}{\sigma_L^m} \right)^2 \right] / ndf \qquad (2)$$

where $ndf$ is the number of degrees of freedom and $m$ is the index of adjacent film pairs
(i.e., [1,2], [2,3], [3,4], …, and [18,19], [19,20] in Fig. 6) or with one skip if there was a
base track inefficiency (i.e., [1,3], [2,4], [3,5], …, [17,19], [18,20]). $\Delta\theta_R^m =$
$(\Delta\theta_x^m \times \tan\theta_x + \Delta\theta_y^m \times \tan\theta_y)/\sqrt{\tan^2\theta_x + \tan^2\theta_y}$ and $\Delta\theta_L^m = (\Delta\theta_y^m \times \tan\theta_x -$
$\Delta\theta_x^m \times \tan\theta_y)/\sqrt{\tan^2\theta_x + \tan^2\theta_y}$, and $\Delta\theta_x^m$ and $\Delta\theta_y^m$ are angular differences along the
$x, y$ coordinates of the ECC. $\sigma_R^m$ and $\sigma_L^m$ are the root-mean-square of $\Delta\theta_R^m$ and $\Delta\theta_L^m$,
which were calculated for every adjacent film pair in every ECC (Fig. 7). Figure 8
shows the distribution of $\chi^2/ndf$ for all tracks in an ECC.
The procedure for track selection is as follows.
1)  Select tracks that start from one of the two most upstream (i.e., summit cone side)

films and stop at one of the two most downstream films.

2)  Select tracks with $\chi^2/ndf < 5.0$.
3)  If a track has any branches, then:

a)  If the shared proportion of track length is ≥20%, choose the longest branch. If

the track lengths are the same, then choose the branch with the smallest

$\chi^2/ndf$.

b)  If the shared proportion of track length is <20% (Fig. 6; case 4), then the

branches were divided into two tracks.

We estimated the effect of the straightness filtering using $\chi^2/ndf < 5.0$. Figure 9 shows
the momentum filtering efficiency. The path length in the lead plates becomes longer
when the track has a larger slope, and thus the momentum also becomes higher. Based
on the background noise study by Nishiyama et al. (2016), the size of the mountain
body used in the simulation and the Izu–Omuroyama scoria cone is broadly the same,
and thus the rejection efficiency should be sufficient. For example, after the track
selection, $1.7 \times 10^6$ tracks were selected at the site "N".




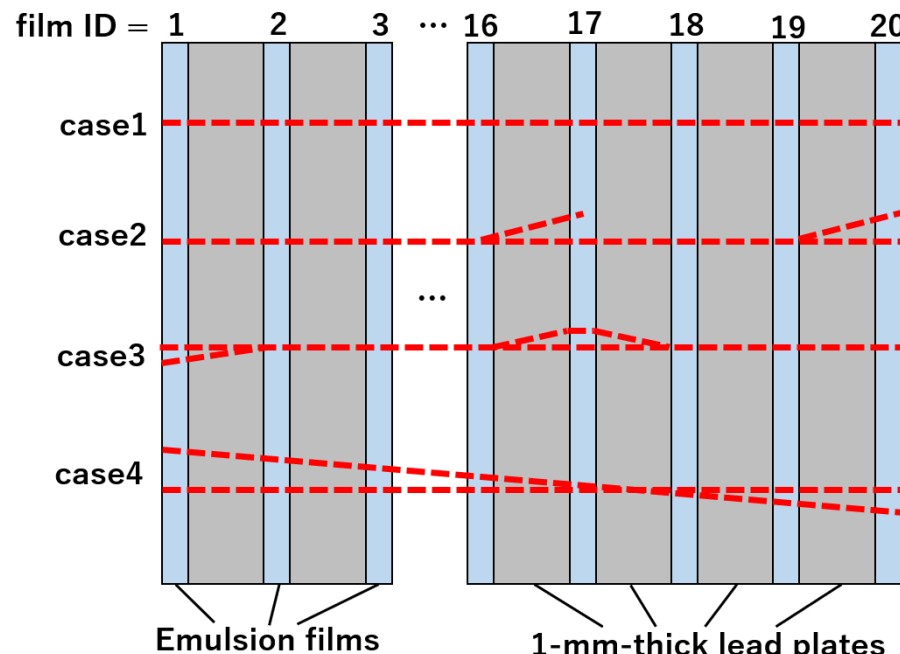



Figure 6. Schematic examples of typical reconstructed tracks in an ECC obtained by

NETSCAN 2.0. Upstream means towards the volcanic cone side and downstream

means the backward free sky direction.

Case 1: a straight track without any branches.

Case 2: a straight track with a branch in the middle and downstream films. The track

branch in the middle was rejected by selection step (1). The branch in the most

downstream film was merged into the straight track by selection step 3a.

Case 3: branches in the upstream and middle films. Both branches were merged into a

straight track by selection step 3a.

Case 4: a pair of straight tracks with small closest distances and similar angles. If the





shared proportion of the track length was <20%, the tracks were divided into two
different tracks by selection step 3b.

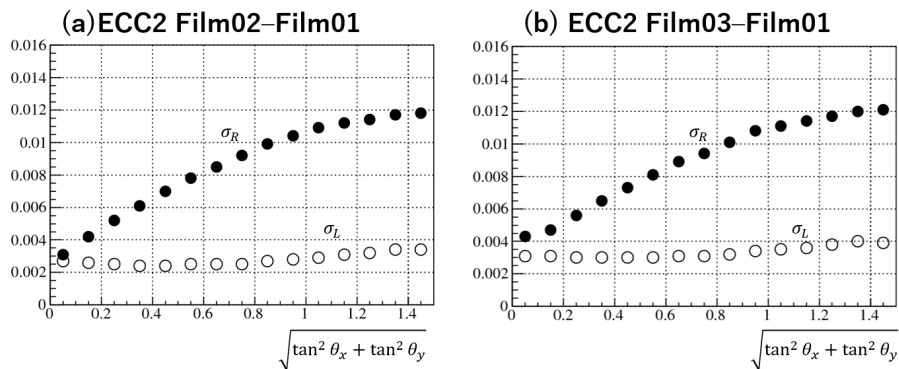


Figure 7. Examples of $\sigma_R$ and $\sigma_L$ as a function of $\sqrt{\tan^2\theta_x + \tan^2\theta_y}$. The values were

determined by the ECC and used to calculate the value of Eq. (2).

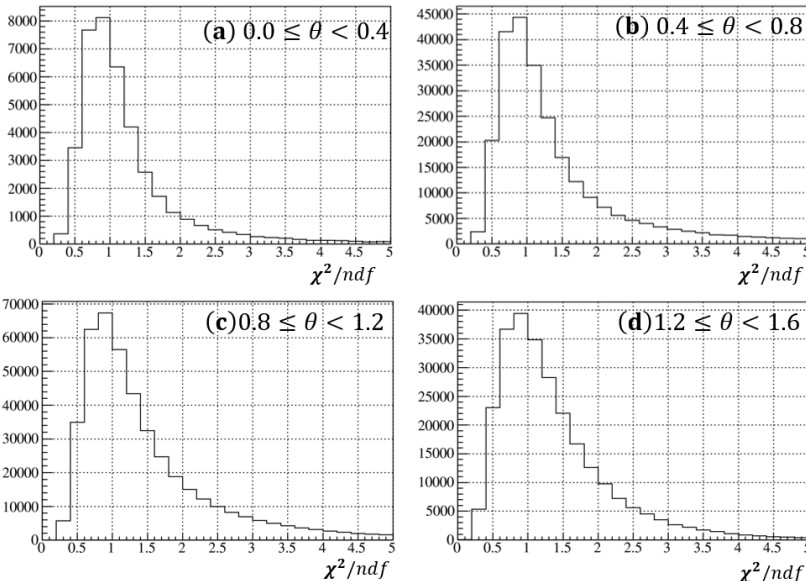


Figure 8. Example of the $\chi^2/ndf$ distribution for selected tracks as a function of $\theta =$

$\sqrt{\tan^2\theta_x + \tan^2\theta_y}$ in an ECC. (a) $0 \leq \theta < 0.4$, (b) $0.4 \leq \theta < 0.8$, (c) $0.8 \leq \theta < 1.2$, and (d)
$1.2 \leq \theta < 1.6$.



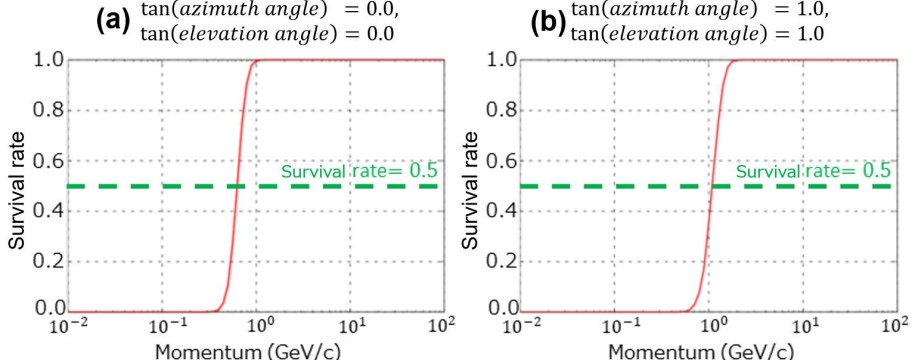


Figure 9. Survival rate of muons after the straightness cut-off as a function of
momentum. (a) Track angles with $\tan(relative\ azimuth) = 0.0$ and
$\tan(elevation\ angle) = 0.0$. (b) Track angles with $\tan(relative\ azimuth) = 1.0$ and
$\tan(elevation\ angle) = 1.0$. The path length in the lead plates becomes longer when the
track has a larger slope, and thus the remaining momentum also becomes higher for
the latter case. The momentum values at a survival rate of 0.5 are 0.6 and 1.1 GeV/c,
respectively.



**4.3 Detection efficiency estimation**
The muon detection efficiency can be estimated by investigating the percentage of
tracks that have a base track in a film. In this paper, we term this percentage the "fill
factor". The fill factor ε can be defined as follows:
$$\varepsilon_j(\theta_x, \theta_y) = \frac{N_j(\theta_x, \theta_y)}{N_{j-1,j+1}(\theta_x, \theta_y)} \qquad (3)$$

where $j$ is a film ID, $N_{j-1,j+1}(\theta_x, \theta_y)$ is the number of tracks in which base tracks were
found in films $j-1$ and $j+1$, and $N_j(\theta_x, \theta_y)$ is the number of tracks in which base
tracks were found in films $j-1, j,$ and $j+1$. The fill factor depends on the films and
track slopes $\theta_x$ and $\theta_y$.
Using the fill factor $\varepsilon_j(\theta_x, \theta_y)$ and $\bar{\varepsilon}_j(\theta_x, \theta_y) = 1 - \varepsilon_j(\theta_x, \theta_y)$, the muon detection
efficiency $\epsilon$ in an ECC can be calculated as follows:
$$\epsilon(\theta_x, \theta_y) = \sum_{hit\ pattern} \varepsilon_1 \times \bar{\varepsilon}_2 \times \varepsilon_3 \times \dots \times \bar{\varepsilon}_{18} \times \varepsilon_{19} \times \varepsilon_{20} \qquad (4)$$

where $hit\ pattern$ is the summation for all possible hit patterns (e.g., $\varepsilon_1 \times \bar{\varepsilon}_2 \times \varepsilon_3 \times$
$\dots \times \bar{\varepsilon}_{18} \times \varepsilon_{19} \times \varepsilon_{20}$ or $\bar{\varepsilon}_1 \times \varepsilon_2 \times \varepsilon_3 \times \dots \times \varepsilon_{18} \times \bar{\varepsilon}_{19} \times \varepsilon_{20}$) from the track selection
conditions described in section 4.2 (Fig. 10). An example of the angular distribution of
the fill factor $\varepsilon_j(\theta_x, \theta_y)$ and muon detection efficiency $\epsilon(\theta_x, \theta_y)$ in an ECC is shown in
Fig. 11. The statistics of observed muons were limited in some angular bins by the
thick volcanic cone. However, the statistics were sufficient in the backward region (i.e.,
elevation angle < 0.0). We used the distribution of the negative elevation angular
region instead of the positive region, because it has enough statistics and the optics of
the HTS has an approximately two-fold rotational symmetry.







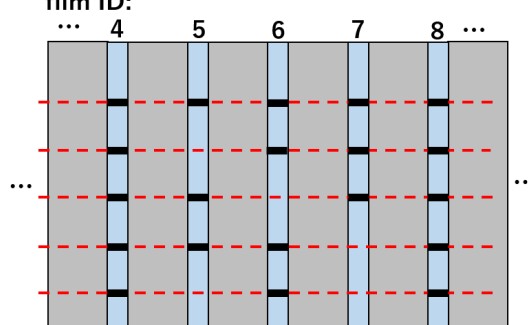


Figure 10. Example of all hit patterns and the products of fill factors in Eq. (4) when

base tracks are found in film ID numbers 4 and 8. The red lines indicate the

reconstructed tracks and the short black lines represent the base tracks found in the

films.







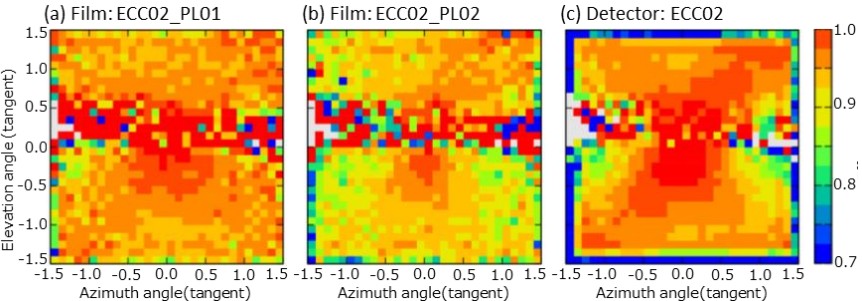


Figure 11. Examples of the angular distribution of the fill factor in some films and the
efficiency of an ECC. (a) Fill factor for film ID = PL01 (most upstream film) and ECC
ID = 02 at site "N". (b) Fill factor for film ID = PL02 and ECC ID = 02. (c) Muon
detection efficiency for ECC ID = 02 as evaluated by Eq. (4). The horizontal axis is the
tangent of azimuth angle; the vertical axis is the tangent of the elevation angle; the
colors represent the fill factor/efficiency values. A positive elevation angle means the
muon path is from the cone; a negative elevation angle means the muon path is from
the backward free sky. The gray color means there were no observed muons in the
angular bin due to the thick volcanic cone.



**5 Results**
The average density along muon path was determined for each observation site. We
used the muon flux model of Honda et al. (2004), energy loss model of Groom et al.
(2001), and topography around the Izu–Omuroyama scoria cone from the Geospatial
Information Authority of Japan (https://maps.gsi.go.jp/). The coordinates of the
observation site, direction, sensitive area, thickness of the ECC detectors, and
observation time were used to calculate the expected number of muons at each
observation site. The expected number of muons can be calculated as a function of the
average density $\rho_k$ along the path:
$$N_k^{simu}(\rho_k) = f_k(\rho_k, L_k) \times S_k \times \Omega_k \times T \times \epsilon_k \qquad (5)$$

where $k$ is the index of an angular bin, $f_k(\rho_k, L_k)$ is the penetrating muon flux
(calculated from the muon flux model, energy loss model, and path length $L_k$), $S_k$ is the
sensitive area of the ECC, $\Omega_k$ is the solid angle, $\epsilon_k$ is the muon detection efficiency, and
$T$ is the observation time.
The angular bin size used for calculating the expected value was $(0.01)^2$ in terms of
the tangent. The angular bins were then merged to improve the statistical accuracy of
the observed values. This merging procedure is useful in topology where a small
change in elevation angle can dramatically change the path length in the volcano. If $k$
is the index of the angular bins of $(0.01)^2$ and the bins belong to a larger angular bin $i$,
then the following equation holds:
$$N_i^{merged}(\rho_i) = \sum_k N_k^{simu}(\rho_i) \quad (6)$$

where $\rho_i$ is the density of the merged angular bin $i$. If $N_i^{obs}$ is the number of the
detected muons in the angular bin $i$, then we can uniquely determine the density value





$\rho_i$, such that $N_i^{merged}(\rho_i) = N_i^{obs}$. The lower limit $\rho_i^{low}$ and upper limit $\rho_i^{up}$ caused by the
statistical error on $N_i^{obs}$ can also be estimated as follows:

$$N_i^{merged}(\rho_i^{low}) = N_i^{obs} + \sqrt{N_i^{obs}} \quad (7)$$


$$N_i^{merged}(\rho_i^{up}) = N_i^{obs} - \sqrt{N_i^{obs}} \quad (8)$$


An example of the derived density map is shown in Fig. 12. All results are shown in
Figs 14–23 (Appendix A).
The definition of the angular bin areas was based on the following. The size of the
angular bins was $(0.2)^2$ when the elevation angle is 0.1 to 0.5 in tangent terms. When
the elevation angle is >0.5, the angular bin size was $(0.1)^2$. If the observed muon count
in the bin was <25, the angular bin was manually merged with adjacent bins to
improve the statistical error. The angular bin with a near-surface path (path length<
30 m) was excluded to avoid ambiguity between the actual topography and digital
elevation map. The attitude errors of each muon detector also contribute to the path
length ambiguity, especially near the surface of the cone.


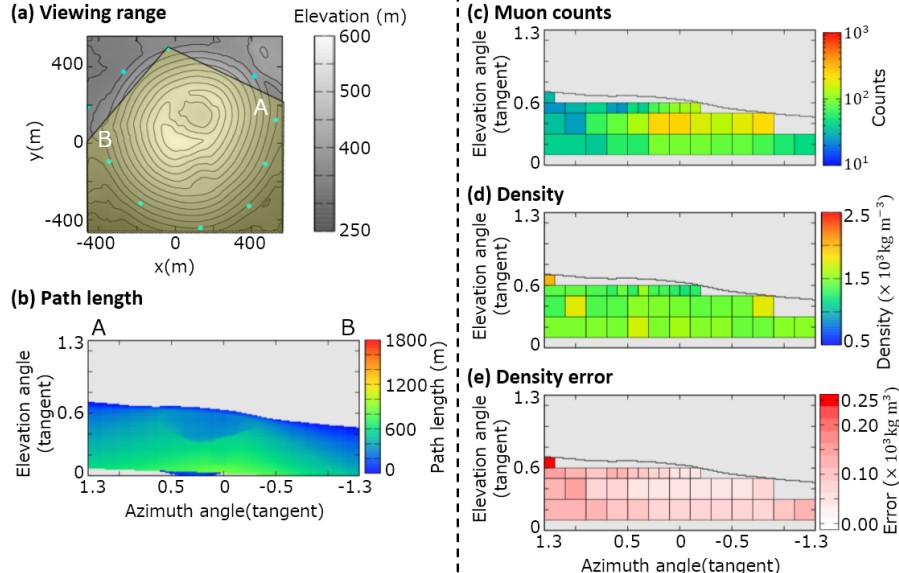



Figure 12. Data for observation site N. (a) Map, topography, and viewing range; (b)

path length of the volcanic cone; (c) muon counts $N_i^{obs}$; (d) density $\rho_i$. The maximum

value of the color bar indicates a density of $>2.5 \times 10^3$ kg m$^{-3}$ and the minimum value

is $<0.5 \times 10^3$ kg m$^{-3}$. (e) Density error $\varDelta\rho = \left(\rho_i^{up} - \rho_i^{low}\right)/2$. The maximum value of the

color bar indicates a density error of $>0.25 \times 10^3$ kg m$^{-3}$.




**6 Validation**
Firstly, we validated the observed muon flux by comparing it with the muon flux
model in the free sky region. The average and standard deviation of the ratio between
the sites were $88\% \pm 4\%$ in the forward direction and $92\% \pm 2\%$ in the backward
direction, except for the NNE site (Fig. 13). There were also 4%–7% in each detector
site except the forward directions at the SE and NNE site (Fig. 14). For reference, a
10% error on the flux corresponds to a 4% error on the density length at a tan(elevation
angle) = 0.2 and density length = 1000 m (water equivalent). These deviations were
less than the errors caused by the muon statistics. The discrepancy for the NNE site is
discusses in the next section.
Secondly, we compared the density of the entire volcanic cone determined by gravity
data with that obtained by muography. Table 2 shows the density determined from
each observation site when the cone is considered to be uniform. The calculation of the
overall density $\bar{\rho}$ is as follows:
$$\bar{\rho} = \frac{\sum_i \rho_i V_i}{\sum_i V_i} \qquad (9)$$
where $i$ is the index of the angular bins and $V_i$ is the volume of the volcanic cone cut off
by the angle bin $i$. Based on the gravity study of Nishiyama et al. (2021), the density of
the Izu–Omuroyama scoria cone is $1.39 \pm 0.07 \times 10^3$ kg m$^{-3}$. The overall density
derived by muography at each observation site is $1.42$–$1.53 \times 10^3$ kg m$^{-3}$, except for
one site. These values are broadly consistent with the density determined from gravity
data, except for the observation site W ($1.72 \times 10^3$ kg m$^{-3}$).





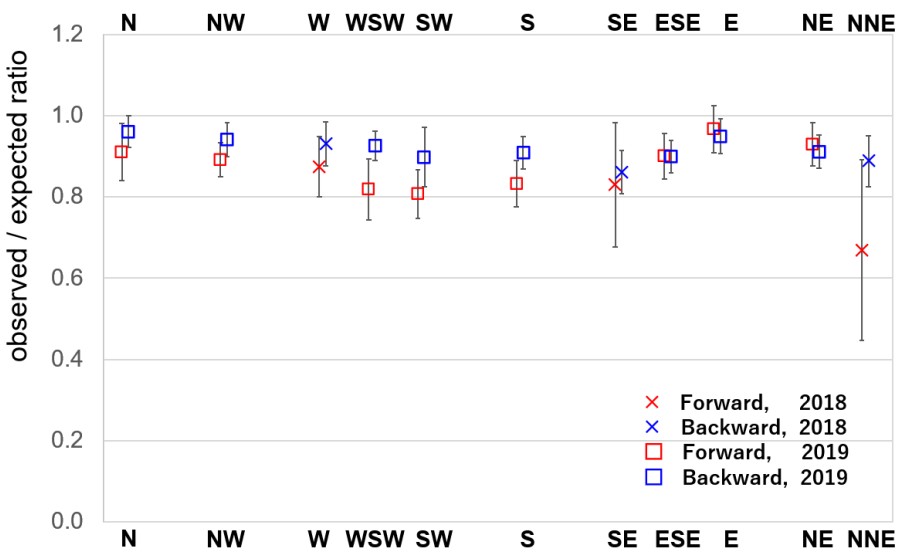


Figure 13. The observed/expected muon flux ratio for each observation site in the

free sky region. The plot represents the average value of the ratio in tangential

angular space, and the error bars are the standard deviations at each site.



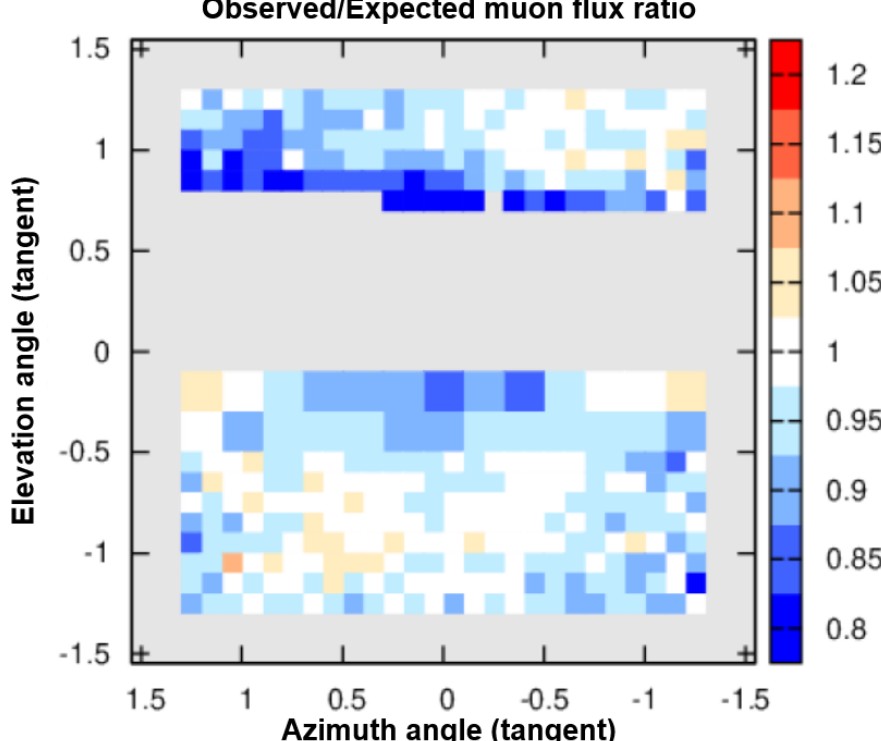


Figure 14. Example of the observed/expected muon flux ratio in the free sky region at
site N. The horizontal axis represents azimuth angle, and the vertical axis represents
the elevation angle. Positive elevation angle means the muons come from forward
directions (the volcanic cone). Negative elevation angle means the muons come from
backward free sky directions. Typical deviation of the ratio is 4%–7% in each site.





| Observation site | Overall density $\bar{\rho}$ ($\times 10^3$ kg m$^{-3}$) | Observation site | Overall density $\bar{\rho}$ ($\times 10^3$ kg m$^{-3}$) |
|---|---|---|---|
| N | 1.51 | S | 1.49 |
| NW | 1.45 | ESE | 1.45 |
| W | 1.72 | E | 1.42 |
| WSW | 1.50 | NE | 1.53 |
| SE | 1.46 | NNE | 1.50 |
| SW | 1.48 | | |

Table 2. Overall bulk density obtained by muography, assuming that the density is
uniform in the volcanic cone.



**7 Discussion**
For the observed/expected muon flux ratio in the free sky region, the values in the
forward direction are less than in the backward direction at many observation sites.
This could be because the detectors were buried in holes on steep slopes (~30°), and our
analysis might not account for that effect. Due to the steep slope, muons arriving from
the forward direction need to penetrate some amount of soil, whereas muons from the
backward direction can enter the detector without being affected by the soil cover. In
addition, the resolution of the detector coordinates is ~3 m, which might also contribute
to the discrepancy.
Some density results from near the ground surface are complex. Some regions near the
path length of 30 m appear to have relatively higher or lower density than the other
data (e.g., Fig. A6, A9). One possible reason for this is the error on the detector
attitude. Near the surface of the volcanic cone, the difference between the calculated
and actual path lengths may become larger due to the error on the detector attitude.
The anomalous data for the NNE site also warrants further consideration. The reason
for this might be a difference between the digital elevation map and actual topography.
There is a stone wall in front of the buried detector at this site, which is about 1 m high
and located on the volcanic cone side. The grid size of the digital elevation map used in
this study is 1 m, and thus the map might not record this steep gradient.
In summary, errors in the position and attitude of the detectors, and the accuracy of
the DEM, might cause a misfit between the DEM and actual topography. These are the
main reasons for the discrepancy between the observed and expected muon flux.
The discrepancy between the observed and expected muon flux was ±4% in the
forward direction and ±2% in the backward direction between the detectors. In





addition, the typical deviation inside each site was 4%–7%. These values are smaller
than the statistical error of the observed muons used to determine the density of the
volcanic cone, and thus they were not significant for our observations. It is interesting
to consider if an improvement in the accuracy of the detector position and attitude, and
the DEM, would decrease this systematic error. For example, the ±4% deviation in the
forward direction would be expected to decrease to ±2%, because the misfit effect is
less in the backward direction. Further improvements will require simulation of the
expected muon flux that take into account more processes and verification of the
systematic errors associated with the ECC detectors.
The obtained density values ($1.42–1.53 \times 10^3 \ \mathrm{kg \, m^{-3}}$; this study) and $1.39 \pm 0.07$
$\times 10^3 \ \mathrm{kg \, m^{-3}}$ (Nishiyama et al., 2021) for Izu–Omuroyama scoria cone are broadly
consistent (Table 2). In a previous study, Rosas-Carbajal et al. (2017) identified an
offset between the density obtained by muon and gravity data and the density obtained
from muon data was $0.5 \times 10^3 \ \mathrm{kg \, m^{-3}}$ less than that obtained from gravity data. In our
validation, this discrepancy does not exist. As Rosas-Carbajal et al (2017) suggested,
the discrepancy might be due to differences in the filtering performance for low-
momentum particles shown in Fig. 9.
The higher density obtained at site W cannot be explained by the systematic errors
described above. One possible reason for this is an actual high-density structure in
front of the site. This hypothesis is consistent with the fact that lava flowed out from
the crater lake to the west (Koyano et al., 1996).



**8 Conclusions**
A muographic study of the Izu–Omuroyama scoria cone was undertaken in 11
directions. The ECC detector design was optimized for quick installation in the field.
We mounted the 11 detectors beneath the ground, surrounding the volcanic cone. The
tracks of charged particles that passed through the ECCs were reconstructed using the
automated emulsion track readout system HTS and NETSCAN 2.0 software. After
track selection, including momentum filtering and efficiency estimation, the density
profiles in 2D angular space were derived for each observation site. The methods
described in this paper can be applied to the observation of other volcanoes and target
objects.
We compared the observed muon flux to the expected value from a muon flux model in
the free sky region. The muon flux difference between each detector was 4% in the
forward directions and 2% in the backward directions, and the typical deviations in
each site were about 4%–7%. The errors on the detector coordinates and attitude, and
DEM, are the main cause of the discrepancy between the observed and expected muon
flux.
In addition, we also compared our results with the overall volcanic cone density
estimated from gravity data, which are broadly consistent, apart from the W site. This
discrepancy for the W site can be explained by the systematic errors discussed in the
previous section and statistical error of the observed muons. It might also reflect a
high-density structure located in the western flank of the volcano. Further 3D density
reconstructions of the Izu–Omuroyama scoria cone are ongoing using the data set
described in this paper.



**Acknowledgements**
The authors thank Hideaki Aoki and his colleagues of Ike-kankou for collaborating on
our study. We also thank Masakazu Ichikawa of the Earthquake Research Institute,
the University of Tokyo, for support during the observation campaign. We are also
grateful for the technical support of the staff and students in F-lab, Nagoya University,
especially with the nuclear emulsion films. This research was supported by JSPS
KAKENHI Grant 19H01988, an Izu Peninsula Geopark Academic Research Grant
(2018), the Joint Research Program of the Institute of Materials and Systems for
Sustainability at Nagoya University (2017–2021), and a JSPS Fellowship (Nagahara;
Grant DC2, 19J13805).



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





**Appendix A**
The density results for each observation site are shown in Figs 15–24.




Figure A1. Observation site NW. (a) Map, topography, and viewing range; (b) path
length of the volcanic cone; (c) muon counts $N_i^{obs}$; (d) density $\rho_i$. The maximum value of
the color bar indicates a density of $>2.5 \times 10^3$ kg m$^{-3}$ and the minimum value is $<0.5$
$\times 10^3$ kg m$^{-3}$. (e) Density error $\Delta\rho = \left(\rho_i^{up} - \rho_i^{low}\right)/2$. The maximum value of the color
bar indicates a density error of $>0.25 \times 10^3$ kg m$^{-3}$.




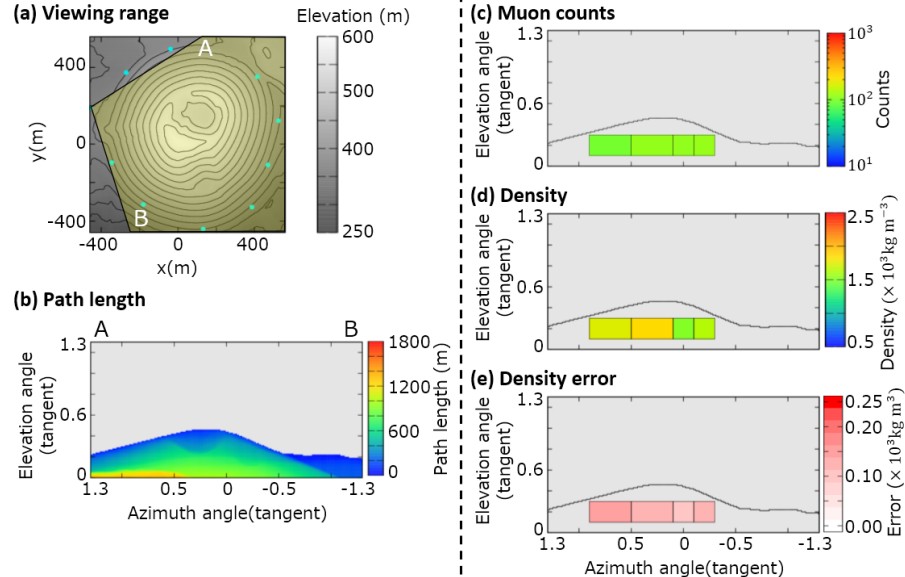



Figure A2. Observation site W. (a) Map, topography, and viewing range; (b) path length
of the volcanic cone; (c) muon counts $N_i^{obs}$; (d) density $\rho_i$. The maximum value of the
color bar indicates a density of $>2.5 \times 10^3$ kg m$^{-3}$ and the minimum value is $<0.5 \times 10^3$
kg m$^{-3}$. (e) Density error $\Delta\rho = \left(\rho_i^{up} - \rho_i^{low}\right)/2$. The maximum value of the color bar
indicates a density error of $>0.25 \times 10^3$ kg m$^{-3}$.




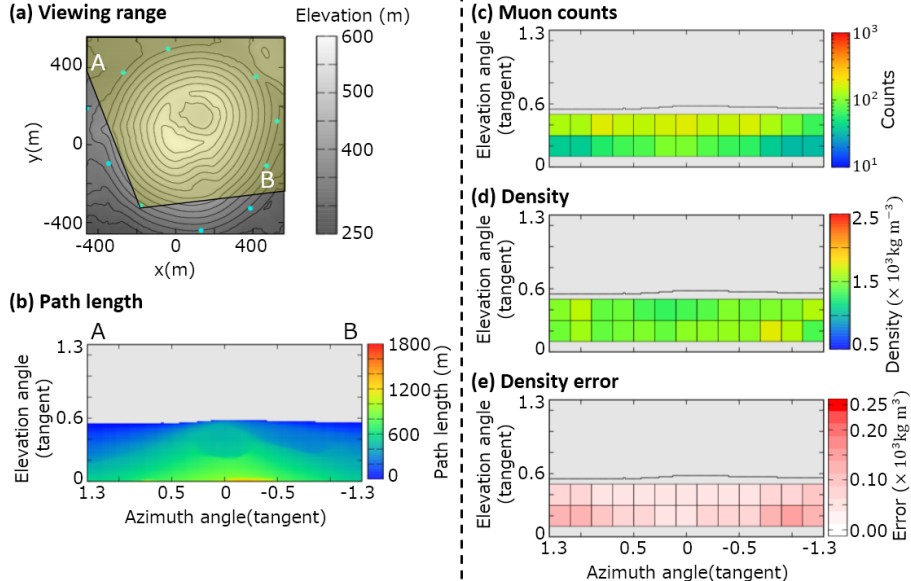



Figure A3. Observation site WSW. (a) Map, topography, and viewing range; (b) path
length of the volcanic cone; (c) muon counts $N_i^{obs}$; (d) density $\rho_i$. The maximum value of
the color bar indicates a density of >2.5 × $10^3$ kg m$^{-3}$ and the minimum value is <0.5
× $10^3$ kg m$^{-3}$. (e) Density error $\Delta\rho = \left(\rho_i^{up} - \rho_i^{low}\right)/2$. The maximum value of the color
bar indicates a density error of >0.25 × $10^3$ kg m$^{-3}$.



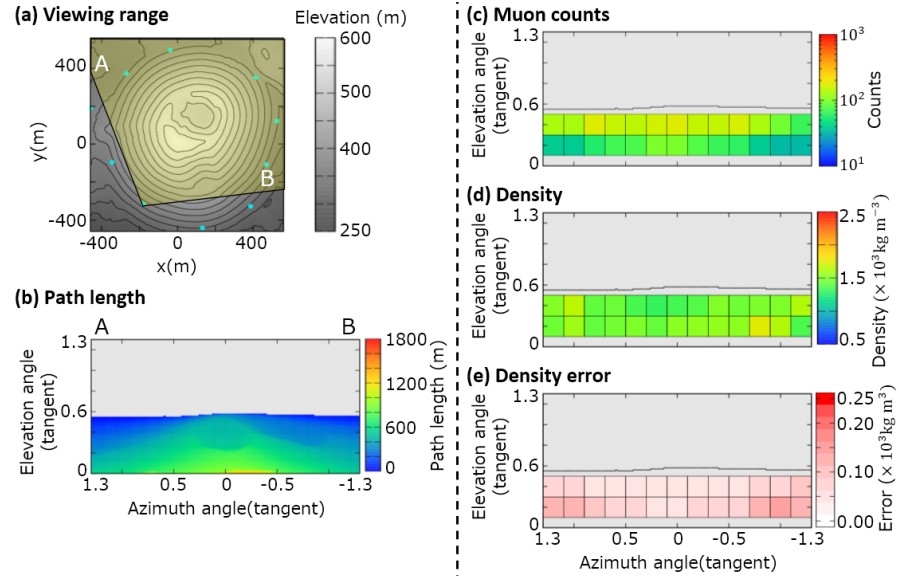


Figure A4. Observation site SW. (a) Map, topography, and viewing range; (b) path
length of the volcanic cone; (c) muon counts $N_i^{obs}$; (d) density $\rho_i$. The maximum value of
the color bar indicates a density of $>2.5 \times 10^3$ kg m$^{-3}$ and the minimum value is $<0.5$
$\times 10^3$ kg m$^{-3}$. (e) Density error $\Delta\rho = \left(\rho_i^{up} - \rho_i^{low}\right)/2$. The maximum value of the color
bar indicates a density error of $>0.25 \times 10^3$ kg m$^{-3}$.







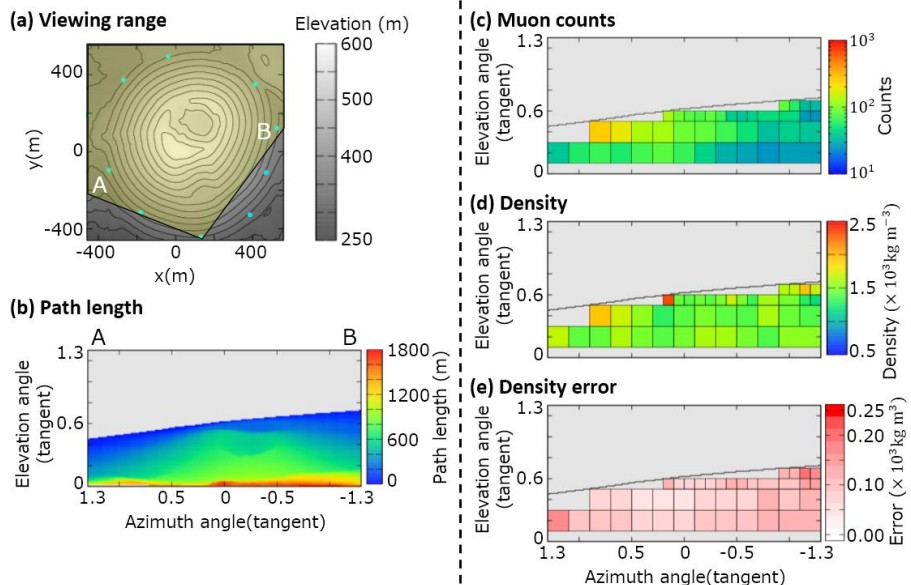



Figure A5. Observation site S. (a) Map, topography, and viewing range; (b) path length
of the volcanic cone; (c) muon counts $N_i^{obs}$; (d) density $\rho_i$. The maximum value of the
color bar indicates a density of $>2.5 \times 10^3$ kg m$^{-3}$ and the minimum value is $<0.5 \times 10^3$
kg m$^{-3}$. (e) Density error $\Delta\rho = \left(\rho_i^{up} - \rho_i^{low}\right)/2$. The maximum value of the color bar
indicates a density error of $>0.25 \times 10^3$ kg m$^{-3}$.





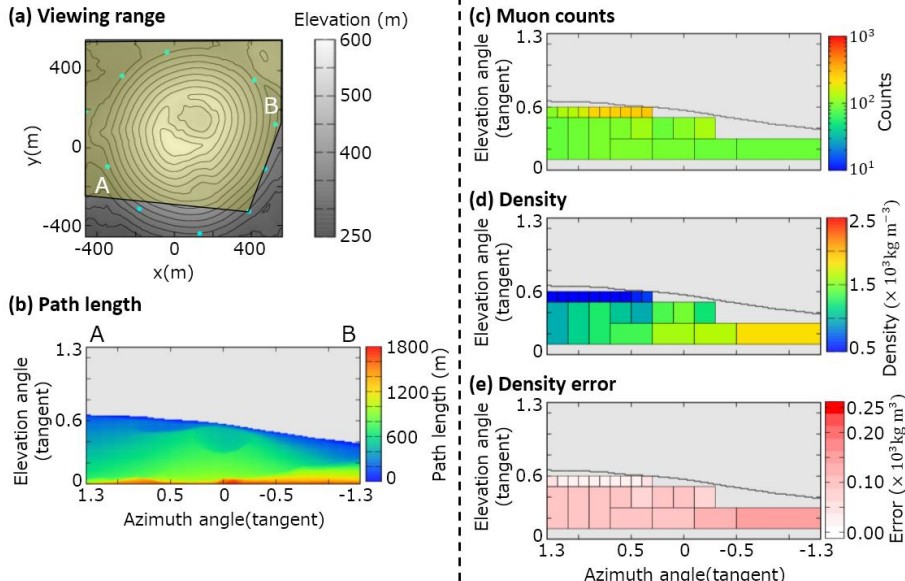



Figure A6. Observation site SE. (a) Map, topography, and viewing range; (b) path
length of the volcanic cone; (c) muon counts $N_i^{obs}$; (d) density $\rho_i$. The maximum value of
the color bar indicates a density of $>2.5 \times 10^3$ kg m$^{-3}$ and the minimum value is $<0.5$
$\times 10^3$ kg m$^{-3}$. (e) Density error $\Delta\rho = \left(\rho_i^{up} - \rho_i^{low}\right)/2$. The maximum value of the color
bar indicates a density error of $>0.25 \times 10^3$ kg m$^{-3}$.






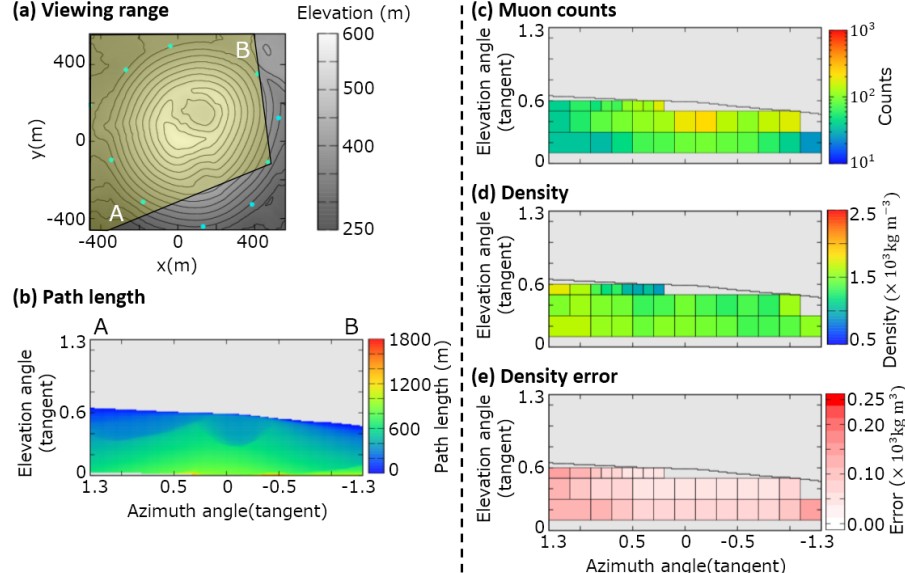



Figure A7. Observation site ESE. (a) Map, topography, and viewing range; (b) path
length of the volcanic cone; (c) muon counts $N_i^{obs}$; (d) density $\rho_i$. The maximum value of
the color bar indicates a density of $>2.5 \times 10^3$ kg m$^{-3}$ and the minimum value is $<0.5$
$\times 10^3$ kg m$^{-3}$. (e) Density error $\Delta\rho = \left(\rho_i^{up} - \rho_i^{low}\right)/2$. The maximum value of the color
bar indicates a density error of $>0.25 \times 10^3$ kg m$^{-3}$.




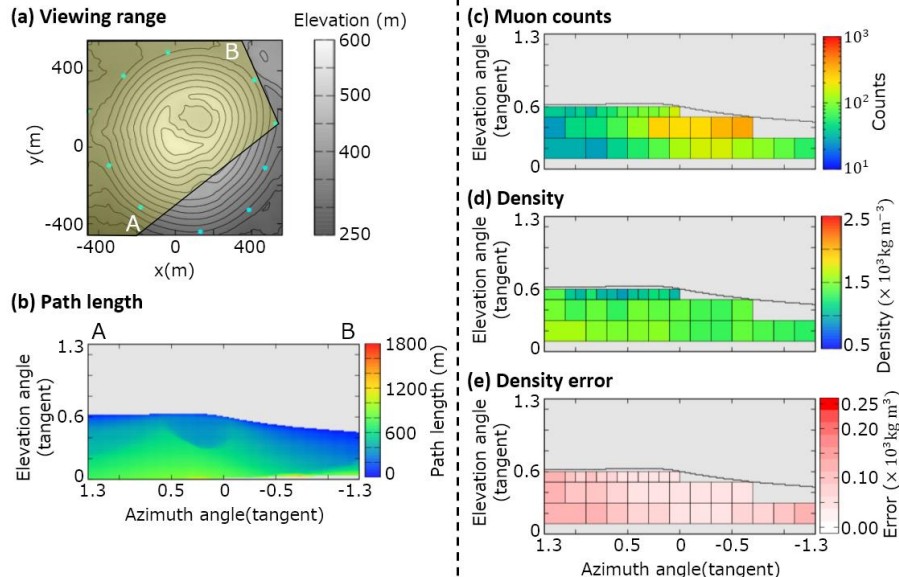



Figure A8. Observation site E. (a) Map, topography, and viewing range; (b) path length
of the volcanic cone; (c) muon counts $N_i^{obs}$; (d) density $\rho_i$. The maximum value of the
color bar indicates a density of $>2.5 \times 10^3$ kg m$^{-3}$ and the minimum value is $<0.5 \times 10^3$
kg m$^{-3}$. (e) Density error $\Delta\rho = \left(\rho_i^{up} - \rho_i^{low}\right)/2$. The maximum value of the color bar
indicates a density error of $>0.25 \times 10^3$ kg m$^{-3}$.






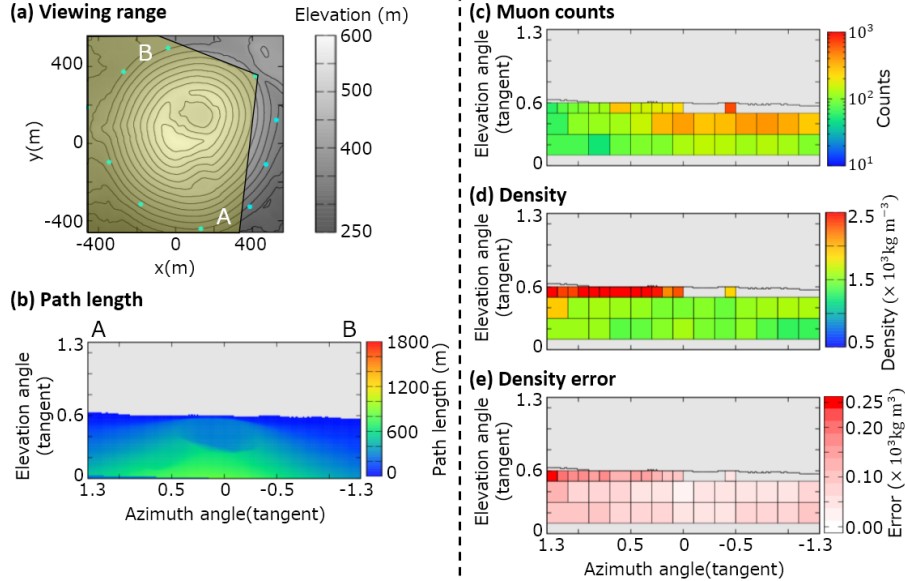



Figure A9. Observation site NE. (a) Map, topography, and viewing range; (b) path length of the volcanic cone; (c) muon counts $N_i^{obs}$; (d) density $\rho_i$. The maximum value of the color bar indicates a density of $>2.5 \times 10^3$ kg m$^{-3}$ and the minimum value is $<0.5 \times 10^3$ kg m$^{-3}$. (e) Density error $\Delta\rho = \left(\rho_i^{up} - \rho_i^{low}\right)/2$. The maximum value of the color bar indicates a density error of $>0.25 \times 10^3$ kg m$^{-3}$.





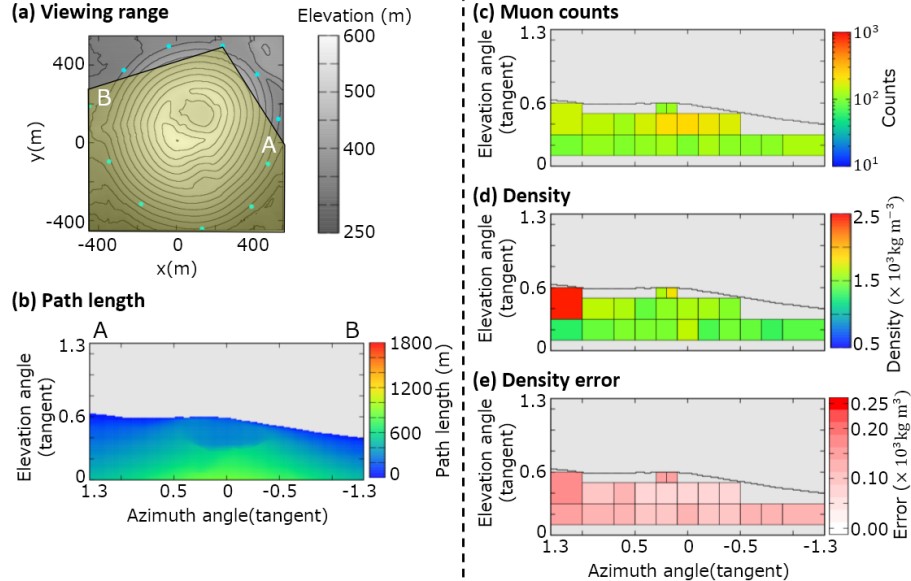



Figure A10. Observation site NNE. (a) Map, topography, and viewing range; (b) path
length of the volcanic cone; (c) muon counts $N_i^{obs}$; (d) density $\rho_i$. The maximum value of
the color bar indicates a density of $>2.5 \times 10^3$ kg m$^{-3}$ and the minimum value is $<0.5$
$\times 10^3$ kg m$^{-3}$. (e) Density error $\Delta\rho = \left(\rho_i^{up} - \rho_i^{low}\right)/2$. The maximum value of the color
bar indicates a density error of $>0.25 \times 10^3$ kg m$^{-3}$.
