# Peer review of "directions using nuclear emulsion cloud chambers"

_Geoscientific Instrumentation, Methods and Data Systems, 2021_

## Author Comment (AC1)

**Appendix A**

The fill factor of the base tracks also depends on the position of the scanned film. The typical causes of the inefficiency are heterogeneous thickness of the emulsion layers, some dusts or scratches on the emulsion surface, and the poorly tuned parameters for the scanning.

Fig. 15 shows the position distribution of the fill factor of all films of an ECC. For example, at upper left the films tend to have the low efficiency (e.g., a-f, h, k, l, q). This part has the larger thickness of emulsion layer because drips were left in the upper left corner when drying after soaking with glycerin solution. Fig. 15(s) and (t) have larger low efficiency area in the right and left. The reason might be the poorly tuned parameters for the scanning.

Compared to the size of the cone, the ECC is a very small "element", thus the local position dependence of the fill factor can be approximately treated as an average fill factor $\varepsilon_j(\theta_x, \theta_y)$. The inefficiency of the basetrack is reflected in the $\varepsilon_j(\theta_x, \theta_y)$ in Eq. (4). Finally, $\varepsilon_j(\theta_x, \theta_y)$, which encompasses the effects of the local inefficiency of the basetrack, is effectively used to derive the angle-dependent muon detection efficiency.

816

817

818

819 Figure 15. The position distribution of the fill factor in each film of ECC02. (a)–(t)

820 represent PL01–PL20, respectively.

821

---

## Author Comment (AC3)

**Appendix A**

We here consider how the position dependency of the detected tracks affects the density
results.

The fill factor (i.e., track detection efficiency in a film) of the base tracks also depends
on the position of the scanned film. The typical causes of the inefficiency are
heterogeneous thickness of the emulsion layers, some dusts or scratches on the emulsion
surface, and the poorly tuned parameters for the scanning.

Fig. 15 shows the position distribution of the fill factor of all films of an ECC. For
example, at upper right the films tend to have the low fill factor (e.g., a-f, h, k, l, q). This
part has the larger thickness of emulsion layer because drips were left in the upper right
corner when drying after soaking with glycerin solution. Fig. 15(s) and (t) have larger
area of low fill factor in the right and left. The reason might be the poorly tuned
parameters for the scanning.

Compared to the size of the scoria cone, the ECC is a very small "element", thus the
local position dependence of the fill factor can be approximately treated as an average
fill factor $\varepsilon_j(\theta_x, \theta_y)$. The inefficiency of the basetrack is reflected in the $\varepsilon_j(\theta_x, \theta_y)$ in Eq.
(4). Finally, $\varepsilon_j(\theta_x, \theta_y)$, which encompasses the effects of the local inefficiency of the
basetrack, is effectively used to derive the angle-dependent muon detection efficiency.

How about the position dependency of noise? Local high density of random silver grains
caused by any chemical conditions, or fake images produced by scratches on the films
might create a group of fake tracks concentrated in one place. Such fake tracks tend to
have small slopes by scanning with automated emulsion readout system. If such noise is
continuous at the same location on the film, they will make many parallel tracks at a
certain slope and give a systematic error in the result. However, such possibility has
been eliminated by the track selection algorithm described in the section 4.2. Because
such concentrated tracks in position and angular space make numerous entangled
linklets. Branches in track connections were removed in the selection. Fig. 16 shows the
number of selected tracks with small slope per mm² in each observation site. There are
no remarkable spikes. The difference of the peak in the histograms depends on the
difference of exposure time (SE, W, NNE), existence of topography in the backward
direction (NE), and pitch angle of the detector attitude (i.e., SW has large pitch angle,
thus less tracks of the small slopes).

[Figure]

Figure 15. The position distribution of the fill factor in each film of ECC02. (a)–(t)

represent PL01–PL20, respectively.

[Figure]

Figure 16. (a) The position distribution of the number of the selected tracks per mm² in the ECC02. (b)–(l) The number of the selected tracks per mm² of the site N–NNE, respectively. These tracks selected for this figure come from in the backward direction, and have small slopes, $|\tan\theta_x| < 0.5$, and $|\tan\theta_y| < 0.5$.

---

## Author Response (AR1)

First of all, thank you for both of referees, for your careful reading of our manuscript and giving us some good comments.
* * *
To the referee 1,

> Lines 93-94: "An ECC detector can measure the momentum of the charged particle by detecting deflection angles caused by multiple Coulomb scattering."â¨
> In my opinion a ECC doesn't measure the momentum. It can provide a statistical information of the momentum of the muons, as correctly described in the section 4.2

 Maybe I've wrote some unclear text.
Although we used the statistical momentum filtering in this study, an ECC detector can measure the momentum of a high-energy charged particle, one by one, as shown in the previous studies (e.g., Agafonova et al., 2012).
I've changed the texts (Line 93-95).
"An ECC detector can measure the momentum of the charged particles, one by one, by detecting deflection angles caused by multiple Coulomb scattering (Agafonova et al., 2012)."

> Line 97: Please describe better  and/or give a reference about the formula (1)

I've added the following reference "Review of Particle Physics", M. Tanabashi et al. (Particle Data Group)
Phys. Rev. D 98, 030001, 2018.
And I also gave the detailed description about some related equations of the reference (Line 95-97).

> Line 117 "at 4 ka"

I've changed   from "at 4 ka" to "about 4,000 years ago".

> Lines 169-170 "we needed to add time information to the ECC"

I've changed the sentences:

Before:
Given that there is no temporal resolution in emulsion films, we needed to add time information to the ECC. In previous muographic studies using emulsion films, researchers have used emulsion films with a different alignment during the muon observations and standby (e.g., Tanaka et al., 2007).

After :
Given that there is no temporal resolution in emulsion films, the ordinary ECC detectors can't distinguish whether the cosmic-ray tracks pass the ECC during muographic observation or transportation and standby. Thus we also add a similar gimmick as previous muographic studies using emulsion films. The researchers have used emulsion films with a different alignment during the muon observations and standby (e.g., Tanaka et al., 2007).

> Line 232: please provide the units of the errors the angles.
> Is  the absolute azimuth  coordinates provided directly by the instrument ?
Yes.
> Which are the angles measured with the FOG and which with the digital leveler ?

I added some additional information to the corresponding texts.
"Measure the attitude of the outer box (i.e., the yaw [absolute azimuth angle], roll, and pitch). The yaw was measured with a fiber optic gyro (Japan Aviation Electronics Industry Ltd.; model FOG JM7711; Watanabe et al., 2000), and roll and pitch were measured by the digital leveler. The typical errors on the yaw, roll, and pitch are $8.7 \times 10^{-3}$, $1.0 \times 10^{-3}$,  and $1.0 \times 10^{-3}$ radians, respectively."

> Lines 310-311: How do you evaluate the filtering efficiency ? It is not described and no reference is given.

I added the following sentence:
This figure was derived from a simple simulation in which the interaction of charged particles inside the ECC was assumed to be multiple Coulomb scattering only, and the scattering angle was approximated by a Gaussian distribution.

> Lines 315-316: how many are the candidate tracks ?

In the end of section 4.1, "... and 1.7 x 10^7 tracks in an entire ECC were reconstructed."

> Lines 444-445: "There were also 4%–7% in each detector  site except the forward directions at the SE and NNE site (Fig. 14)."
> This sentence is not clear to me.

I've modified.

Before:
There were also 4%–7% in each detector  site except the forward directions at the SE and NNE site (Fig. 14).

After:
An example of observed/expected muon flux ratio angular distribution of the site N is shown in Fig. 14. As can be seen in this figure, in each detector site, the inhomogeneous distribution of the observed/expected muon flux ratio exists. The deviations were 4%–7% except the forward directions at the site SE and NNE.
* * *
To the referee 2,

> There is only one aspect that needs a clarification. ...........
> ..... the text does not clarify enough the treatment of position-dependent performance variations of the emulsion films.

I've added a new appendix and you can see the figures (Fig. 15-16) of position distribution of fill factor and the number of selected tracks per mm$^2$ there.